# The role of direct air capture in achieving climate-neutral aviation

Nicoletta Brazzola [1] ✉, Amir Meskaldji[1], Anthony Patt [1], Tim Tröndle [1] & Christian Moretti [1,2]

Growing demand for air travel and limited scalable solutions pose significant challenges to the mitigation of aviation's climate change impact. Direct air capture (DAC) may gain prominence due to its versatile applications for either carbon removal (direct air carbon capture and storage, DACCS) or synthetic fuel production (direct air carbon capture and utilization, DACCU). Through a comprehensive and time-dynamic techno-economic assessment, we explore the conditions for synthetic fuels from DACCU to become cost-competitive with an emit-and-remove strategy based on DACCS under 2050 $CO_2$ and climate neutrality targets. We find that synthetic fuels could achieve climate neutrality at lower cost than an emit-and-remove strategy due to their ability to cost-effectively mitigate contrails. Under demand reductions, contrail avoidance, and $CO_2$ neutrality targets the cost advantage of synthetic fuels weakens or disappears. Low electricity cost (€0.02 kWh$^{-1}$) and high fossil kerosene prices (€0.9 l$^{-1}$) can favor synthetic fuels' cost-competitiveness even under these conditions. Strategic interventions, such as optimal siting and the elimination of fossil fuel subsidies, can thus favor a shift away from fossil-reliant aviation.

Aviation has historically contributed to approximately 4% of anthropogenic climate warming[1]. About two-thirds of aviation's radiative forcing is attributed to non-$CO_2$ effects, such as contrail cirrus cloud formation or indirect effects due to nitrous oxide emissions[2–5]. While aviation's historical contribution to climate change may appear small, its role in the future could be significant due to the expected growth of the sector and the challenges of mitigating its emissions[6–11]. The effects of viable decarbonization options, such as operational improvements and efficiency gains, are currently jeopardized by rising demand[12–14], and the switch to biofuels is constrained by biophysical limits, such as the availability of sustainable biomass, which is also in demand for other mitigation purposes[15–17]. While some mitigation technologies, such as hydrogen and electric aircraft, could theoretically curb all direct flight emissions, they are not yet technically feasible, especially for long-haul flights, and would require a complete renewal of the global aviation fleet[18–21].

This led to the emergence of two potentially scalable climate mitigation strategies: offsetting aviation emissions with carbon dioxide removals (CDR)[22–26] and deploying renewable Fischer-Tropsch synthetic fuels from air-captured $CO_2$ and green hydrogen[12,27–29]. To ensure scalability, both solutions could rely on direct air capture (DAC), as this technology has relatively small land and water footprints and does not require biomass[17,30–33]. DAC can be used either in combination with $CO_2$ storage to offset aviation emissions (as direct air carbon capture and storage [DACCS]) or to produce synthetic fuels via Fischer-Tropsch synthesis (as direct air carbon capture and utilization [DACCU]). In addition to its potential for scalability, especially if deployed in remote areas[31,34], the use of DAC to tackle aviation's climate impacts could benefit climate mitigation in a larger sense; bearing the high initial costs of this technology can be seen as an equitable strategy[35] to overcome the steepest segment of its learning curve[36–39] and realize its economic viability for other applications. Financing improvements in DAC via increases in ticket prices would indeed fall most heavily on middle-to-high income consumers and households[40,41] but provide long-term benefits for the entire world by making the technology ready for large-scale CDR[37,38,42], which will be necessary to remedy overshoots of a Paris-aligned carbon budget[43,44].

[1]Institute for Environmental Decisions, ETH Zürich, 8092 Zürich, Switzerland. [2]Laboratory for Energy Systems Analysis, PSI Center for Energy and Environmental Sciences, 5232 Villigen, Switzerland. ✉e-mail: Nicoletta.brazzola@usys.ethz.ch

**Table 1 | Input parameters and assumptions underlying all scenarios considered**

| Parameter | Value | Reference |
|---|---|---|
| Growth aviation demand* | +2% yearly | 1,8,10,85–88 |
| Fuel efficiency increase* | +2% yearly | Based on ICAO's target[69] |
| Learning rate DAC* | 12% | 39 |
| Learning rate electrolysers* | 8% | Based on average future CAPEX estimates from[102–105] |
| Learning rate CO reduction* | 7.5% | 106 |
| Price of fossil kerosene* | €0.6 l$^{-1}$ | Average price between 2021-2023[107] |
| Levelized cost of electricity* | 0.03 € kWh$^{-1}$ | Corresponding to 2022 onshore wind cost[71] |
| Discounting | 0% | To ensure intergenerational equity following Emmerling et al. [108] |
| Low-temperature heat | 10 € kWh$_{th}^{-1}$ | 109,110 |
| High-temperature heat | 40 € kWh$_{th}^{-1}$ | 109,110 |
| $CO_2$ transport and storage cost | 20 € tCO$_2^{-1}$ | 109 |

Starred parameters indicate values on which a sensitivity analysis was performed.

On this background, we explore the use of DAC for medium-term mitigation of the aviation sector's climate impacts and investigate the conditions under which the use of DACCU-based synthetic fuels could be more cost-effective than offsets via DACCS. Previous techno-economic assessments have concluded that DACCS is a more cost-effective option for achieving $CO_2$-neutral aviation globally[22,45]. However, they also noted that these cost benefits may not materialize because they are based on uncertain assumptions[45] and that DACCS offers fewer co-benefits, such as potential mitigation of non-$CO_2$ impacts[2,46,47] and alignment with fossil fuel phase-outs[45]. The only study that compared the deployment of DACCS and DACCU to achieve climate neutrality concluded that it is unrealistic to rely entirely on DACCU-based fuels for European aviation fuel consumption if green hydrogen production is to take place only in Europe[26].

In this study, we aim to broaden the discussion by offering a global perspective on DAC deployment to achieve $CO_2$ and climate neutrality in aviation. The global focus is justified by emerging trends in countries such as Chile, Saudi Arabia, Australia, and Morocco, which are positioning themselves as producers of cheap renewable energy and exporters of green hydrogen thanks to their abundant land and renewable energy resources[48,49]. In addition, recognizing the imperative to emancipate aviation from fossil entanglements[50], the societal preferences for DACCU over DACCS[51] and, more generally, the priority of direct emissions reductions over removals[52–54], we set out to identify the conditions under which DACCU can become cost-competitive with DACCS and even with a 'business-as-usual' scenario. By examining the drivers of future costs and policy implications, we present a comprehensive analysis that provides decision-makers with actionable insights to enable DACCU to take off.

## Results
### Scenarios and framework
Although a portfolio of solutions will likely be employed to address the climate impact of aviation, this study exclusively focuses on the role of DAC in achieving $CO_2$ and climate neutrality in the global aviation sector by 2050. We selected 2050 as the target year since numerous national net-zero emissions targets are set for that year[55], including those for the aviation sector[56]. Additionally, we tested whether the results would differ for a net-zero target by 2060 (Supplementary Fig. 10). In the 'DACCU' scenario, synthetic fuels produced from green hydrogen and $CO_2$ captured by DAC lead to a gradual substitution of fossil fuels, eventually replacing conventional jet fuels entirely by 2050. This substitution follows a power curve that we consider more realistic than a typical S-shape diffusion[57–61] due to the extremely ambitious ramp-up of synthetic fuels required by 2050. Conversely, the 'DACCS' scenario focuses on the incremental DACCS-based offsetting of continued fossil jet fuel use. To

ensure comparability, the share of emission offsets follows the same curve of DACCU deployment, reaching 100% by 2050. Finally, we also consider a 'business-as-usual' scenario, in which the aviation sector exclusively relies on fossil jet fuels (kerosene), with no supplementary mitigation measures. The default assumptions underlying all scenarios are that the aviation sector experiences growth at a rate of 2% annually, efficiency improvements leading to a 2% yearly reduction in fuel intensity, a constant levelized cost of electricity aligned with that of onshore wind (€0.03 kWh$^{-1}$), and a constant fossil kerosene price (€0.6 l$^{-1}$) (Table 1 and Methods). This comprehensive framework enables a holistic comparison of DACCU, DACCS and conventional aviation based on fossil kerosene in terms of costs, energy use, and climate impacts.

Our analysis includes two different 2050 goals for the aviation sector. The first is to achieve $CO_2$ neutrality, that is, to reduce $CO_2$ emissions to net-zero by 2050. In the DACCS pathway, this means offsetting $CO_2$ emissions only. In the DACCU pathway, fuel substitution is assumed to fully eliminate $CO_2$ emissions (except for indirect emissions, Methods). Since DACCU-based fuels are expected to burn cleaner[46,47], this pathway also achieves a partial mitigation of the non-$CO_2$ effects (Supplementary Fig. 1). Therefore, the climate benefits of the two pathways are not equal under a $CO_2$ neutrality target. The second target, climate neutrality, on the other hand, includes non-$CO_2$ effects and thus enables a more balanced comparison of the two technology pathways. In fact, to achieve climate neutrality both pathways must neutralize any residual non-$CO_2$ effect with the deployment of DACCS. Figure 1 shows the breakdown in flight emissions, indirect emissions, and removals via DACCS of the different pathways and of a business-as-usual with fossil kerosene to achieve the different targets.

By integrating different technology and climate target scenarios, our framework enables a holistic comparison of DACCU, DACCS and conventional aviation based on fossil kerosene in terms of costs, energy use, and climate impacts.

### The specific climate target determines which option is cheaper
We first calculate the costs per liter fuel in 2050 of the two technology pathways to achieve $CO_2$ and climate neutrality under our standard input assumptions (see Table 1, Methods and Supplementary Tables 1–3). For $CO_2$ neutrality, the DACCS pathway is significantly less costly than the DACCU pathway, which it outperforms by about €0.3 l$^{-1}$ in 2050 (Fig. 2). A postponement of the $CO_2$ neutrality target to 2060 results in a diminished advantage for DACCS (€0.12 l$^{-1}$) due to an increase in the total volumes of $CO_2$ emissions that it needs to offset (Supplementary Fig. 10). The cost penalty of DACCU is mainly due to the high electricity and capital costs of electrolysis essential for synthetic fuel production. The cost comparison under $CO_2$ neutrality does not capture the full benefits of DACCU-based fuels because the reduction in

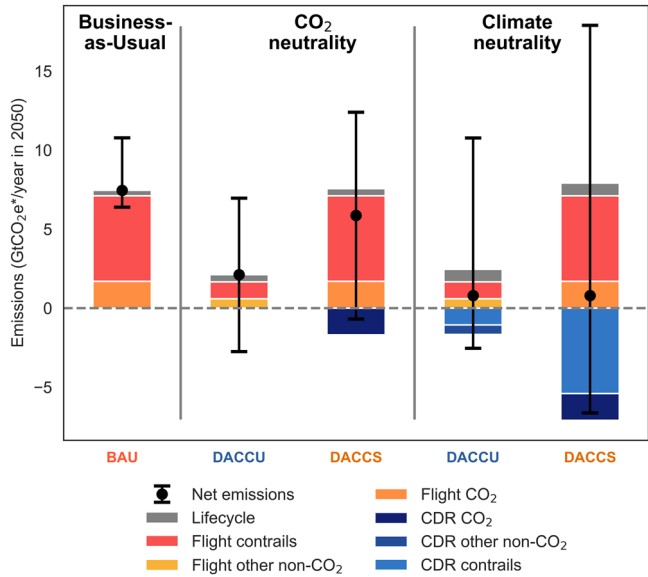

**Fig. 1 | Overview of emissions and removals in gigatonne CO₂-equivalent emissions based on the modified global warming potential (GWP*) metric.** The emit-and-remove scenario based on Direct Air Carbon Capture and Storage ("DACCS" scenario) continues to involve the combustion of fossil kerosene, which results in both CO₂ and non-CO₂ climate impacts, including those associated with condensation trails (contrails). To achieve either CO₂ or climate neutrality, this scenario employs DACCS to offset flight emissions. In contrast, the Direct Air Carbon Capture and Utilization ("DACCU") scenario involves a gradual replacement of fossil kerosene with DACCU-based synthetic fuels, with the aim of achieving complete substitution by 2050. These fuels directly eliminate CO₂ emissions from flights and reduce non-CO₂ impacts (Supplementary Fig. 1 for a breakdown of the radiative forcing effect of DACCU-based emissions). Additionally, the DACCU scenario additionally employs CDR under climate neutrality to neutralize its residual non-CO₂ impacts. The black bars represent the uncertainty of net aviation emissions (the black dots), which results from uncertainties in the non-CO₂ impacts of aviation.

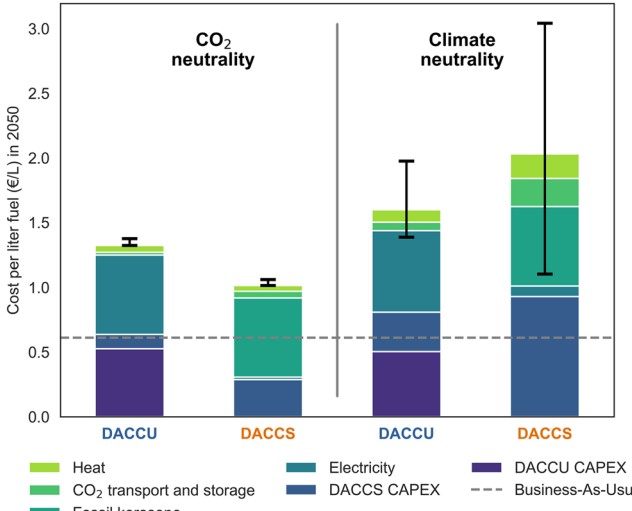

**Fig. 2 | Cost per liter fuel to achieve CO₂ and climate neutrality in the year 2050.** Cost are calculate for a scenario where synthetic fuels replace 100% of kerosene by 2050 via Direct Air Carbon Capture and Utilization ("DACCU") and for a scenario where fossil kerosene is used continuously, and emissions are offset through Direct Air Carbon Capture and Storage ("DACCS"). Black bars represent the uncertainty deriving from the non-CO₂ impacts of aviation.

non-CO₂ impacts, and particularly contrails (Fig. 1), due to cleaner synthetic fuels is not reflected in the cost (Supplementary Fig. 1). Both pathways result in substantially higher costs than a business-as-usual scenario, with DACCU constituting about a doubling of the fuel cost.

Attaining climate neutrality demands a more substantial effort, as it increases the cost per liter of fuel by about a factor of three relative to the business-as-usual scenario (Fig. 2), with cumulative additional costs exceeding 13 trillion euros (Supplementary Fig. 9). Under climate neutrality, where the net climate impacts of the two pathways are identical, the DACCU pathway has a cost advantage over DACCS, which it outperforms by over €0.4 l[-1] in 2050. The higher cost of the DACCS pathway is attributable mainly to the higher carbon removal rates required to offset non-CO₂ emissions, which are higher than in the DACCU pathway (Supplementary Figs. 1, 2). The large offset requirements are due to the sustained demand growth assumed in the analysis. However, assuming no growth of the sector still results in a competitive advantage of the DACCU pathway. Despite its economic advantage, the DACCU pathway results in higher electricity consumption due to energy-intensive electrolysis (Supplementary Fig. 3). This limits its scaling potential to regions with abundant and affordable renewable energy. Finally, both DACCS and DACCU pathways are more expensive alternatives compared to the continued use of fossil kerosene, highlighting the role of policy interventions to propel these pathways forward.

## DACCS has higher costs per emission but is more efficient in scaling DAC

Looking at the total costs for abated emissions relative to the business-as-usual (Fig. 3a), the resulting picture is almost opposite than the one

drawn when looking at cost per liter, or absolute costs (Supplementary Fig. 8c). Under the CO₂ neutrality target, the DACCS pathway has the highest costs per emissions abated, reaching abatement costs of over €500 tCO₂e*[-1] compared to less than €200 tCO₂e*[-1] for the DACCU pathway. This difference arises because DACCS only includes costs associated with reducing CO₂ emissions. Conversely, in the DACCU pathway, the abatement extends to non-CO₂ emissions, thereby increasing the total volume of abated emissions over which the costs are distributed. In the context of climate neutrality, where both technology pathways result in the same level of emissions reduction, DACCU once more emerges as the more cost-effective option due to the reduced quantities of CDR to offset the residual non-CO₂ effects. However, the difference in cost is considerably less pronounced in comparison to the CO₂-neutrality target.

Apart from mitigating the aviation sector, both options could also serve as means of scaling up DAC[62]. This rationale is rooted in the potential role that the aviation sector could play as a niche for the initial deployment of DAC, as the sector is bound to face significant costs in mitigating its emissions due to the lack of affordable alternatives. This perspective results in a picture opposite to that of cost-effective abatement. We find that, due to its higher total DAC capacity (Supplementary Fig. 2), the DACCS pathway consistently offers a lower cost per DAC unit than the DACCU pathway (Fig. 3b). DACCU incurs higher costs due to the production of green hydrogen. This has a significant impact on the cost per unit of DAC installed.

## The cost difference for a CO₂-neutral flight with DACCS and DACCU is small

We further assess the increase in cost per flight per passenger to achieve CO₂ and climate neutrality via the DACCS and DACCU pathways (Fig. 4a). Relying exclusively on DAC-based mitigation results in an increase in costs per passenger flight between €23-260 to achieve CO₂ neutrality and between €35-410 to achieve climate neutrality. In the context of CO₂ neutrality, offsetting aviation CO₂ emissions with DACCS proves to be more economical than fueling the same flight with DACCU-based synthetic fuels. However, the cost difference per passenger is modest, ranging from approximately €20-60 for long-haul flights (London-New York and London-Perth) to only €7 for a short-haul flight from London to Berlin. While the overall cost per passenger

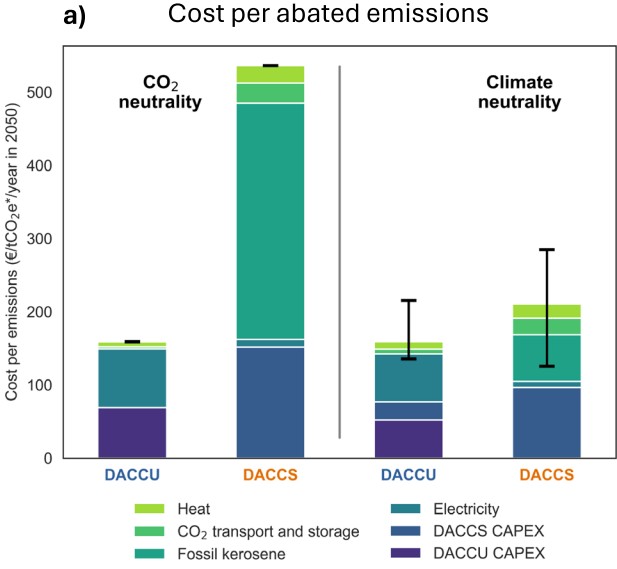
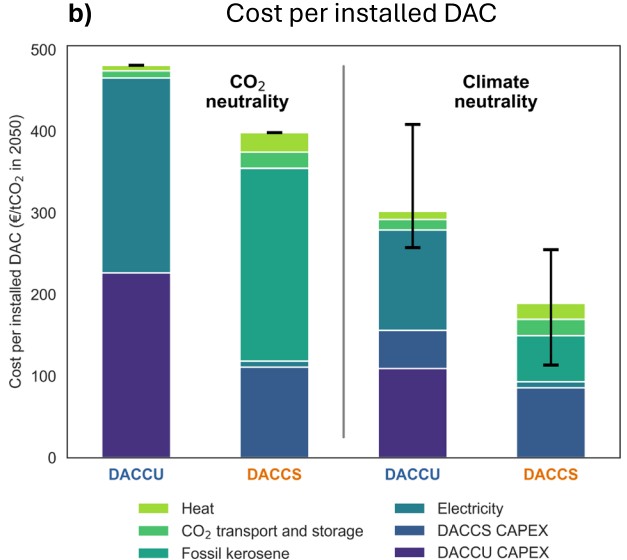

**Fig. 3 | Cost of achieving CO₂ and climate neutrality by 2050.** **a** divided by abated emissions and **b** divided by the installed units of DAC. Costs are shown for a scenario where synthetic fuels replace 100% of kerosene by 2050 (Direct Air Carbon Capture and Utilization, "DACCU") and for a scenario where fossil kerosene continues to be used and emissions are offset by Direct Air Carbon Capture and Storage ("DACCS"). Black bars represent the uncertainty deriving from the non-CO₂ impacts of aviation.

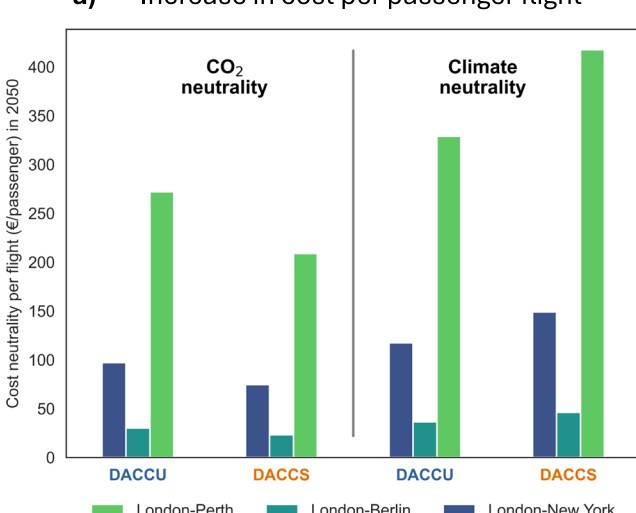
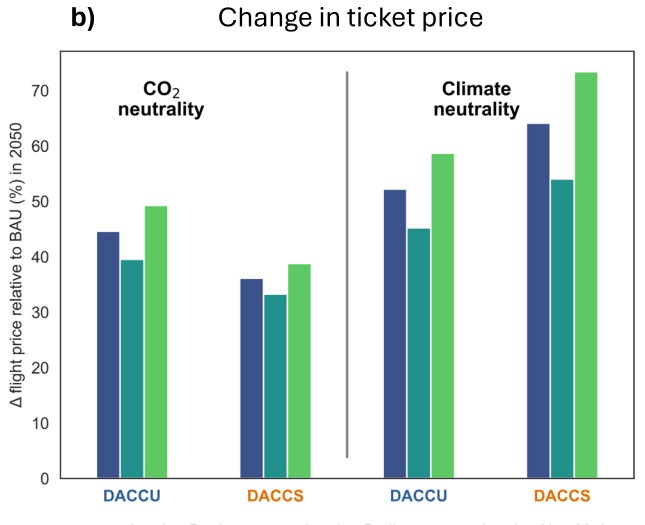

**Fig. 4 | Cost per flight per passenger due to Direct Air Capture-based aviation.** **a** Total costs per flight per passenger and **b** change cost per flight per passenger relative to business as usual to achieve either CO₂ or climate neutrality in 2050 for representative short-, medium-, and long-haul flights under the Direct Air Carbon Capture and Utilization ("DACCU") and Direct Air Carbon Capture and Storage ("DACCS") pathways.

increases to achieve climate neutrality, DACCU becomes cheaper than DACCS, saving about €30-90 per passenger on long-haul flights and €10 on short-haul flights.

We also assessed the impact on the cost of flying relative to the expected future cost of flying in a business-as-usual scenario with continued use of fossil kerosene (Fig. 4b). The projected increase in ticket prices for flights in 2050 ranges between approximately 40-50% for DACCU and 30-40% for DACCS to achieve CO₂ neutrality, rising to up to 60% (DACCU) and 75% (DACCS) to achieve climate neutrality. However, the increase in price is not the same for all flights, since the contribution of fuel costs to ticket prices varies for different routes, as the price is adjusted to demand and to endure competition. While the increases in price due to a complete neutralization of the climate effects of a flight may seem substantial, they lie well below the range of current variance in prices. Indeed, the difference in price between

buying a ticket two weeks or two months in advance is, on average, 400% for the London-Berlin route, over 100% for the London-New York route, and 70% for the London-Perth route[63].

**Additional mitigation options make DACCU less cost attractive than DACCS**

The key advantage of DACCU lies in its ability to neutralize non-CO₂ effects, in particular contrails, at reduced cost, at lower costs than DACCS, which faces rising costs as emissions increase. In reality, however, some cost-effective solutions to mitigate contrails and short-lived non-CO₂ effects are emerging, such as constraining demand growth[24,25], rerouting flights to avoid contrail formation[8,64], and hydrogenating fossil kerosene to reduce aromatics-induced contrail seeding[65,66]. Furthermore, the additional expense associated with a transition to a 100% DAC-centric aviation sector is likely to result in a

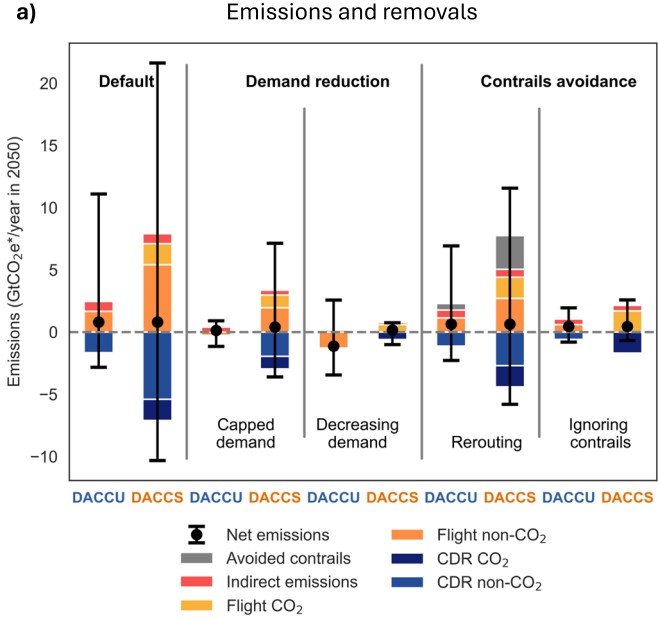

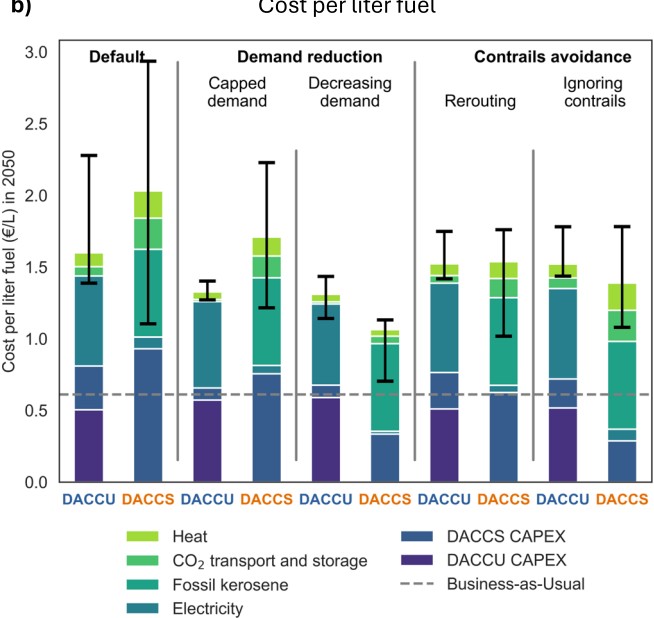

**Fig. 5 | Additional mitigation due to demand changes and contrail mitigation.**
Emissions (**a**) and cost per liter fuel (**b**) to achieve climate neutrality by 2050 for a
scenario where synthetic fuels replace 100% of kerosene by 2050 (Direct Air Carbon
Capture and Utilization, "DACCU") and for a scenario where fossil kerosene con-
tinues to be used and emissions are offset by Direct Air Capture and Storage
("DACCS"). Additionally to the "Default" scenario, which applies the standard
assumptions shown in Table 1, two additional set of measures are shown: (i)
demand reduction measures, whereby the "Capped demand" scenario applies a
zero-growth in demand after 2020, and the "Decreasing demand" experiences a -2%
yearly decrease in demand; and (ii) contrails avoidance measures, whereby
"Rerouting" avoids 50% of contrails at an increase in fuel by 1% and "Ignoring
contrails" eliminates all contrails while ignoring additional mitigation costs. Black
bars represent the uncertainty deriving from the non-$CO_2$ impacts of aviation.

reduction in demand, given the price elasticity of demand[67,68]. While
our modeling does not explicitly include the impact on demand of
DAC-based mitigation, we examine here four additional scenarios for
achieving climate neutrality that explore these demand-side and con-
trail mitigation options, namely: (1) capping demand at 2020 levels,

stabilizing contrail radiative forcing ("Capped demand"); (2) reducing
demand by 2% annually from 2020, reducing contrail radiative forcing
("Decreasing demand"); (3) rerouting flights, which reduces contrails
by 50% with a 1% increase in fuel consumption[8] ("Rerouting"); and (4)
ignoring contrails by assuming they are mitigated in some other way
while excluding the cost of this additional mitigation ("Ignoring
Contrails").

All these options reduce emissions compared to the "Default"
scenario, which is the climate neutrality scenario assessed so far
(Fig. 5a). The "Decreasing demand" scenario achieves the biggest
reductions. By combining DAC-based fuels with demand reductions,
the DACCU scenario goes beyond climate neutrality and eliminates its
reliance on CDR. The "Rerouting" scenario results in the highest
absolute emissions, but these are still significantly lower than those in
the default DACCS scenario.

Overall, the additional measures reduce the cost of achieving
climate neutrality (Fig. 5b and Supplementary Fig. 9). In the "Capped
demand" scenario, DACCU retains a slight cost-advantage because the
constant demand for aviation and the emission reductions of DACCU
eliminate the need for CDR. If demand is not only stagnant, but actually
declines ("Decreasing demand" scenario), the DACCS pathway also
eliminates the need for CDR because of the reduced contrail forcing,
resulting in the lowest cost per liter. However, growth rates have his-
torically been close to +4%, and enforcing international caps may be
challenging because of the lack the necessary governance
frameworks[13,69]. If demand continues to grow at +4%, DACCU would
achieve climate neutrality at a significantly lower cost than DACCS
(Supplementary Figs. 7, 8). Tackling contrails rather than demand may
therefore be a more realistic option. "Rerouting" equalizes the costs of
the two DAC-based pathways, showing that the cost advantage of
DACCU is sensitive to the magnitude of contrail forcing, which is
uncertain[5]. Finally, considering all non-$CO_2$ species other than contrails
("Ignoring contrails") favors DACCS. This is due to the fact that
switching to DACCU-based fuels also reduces non-$CO_2$ species such as
$SO_4$ emissions, which have a cooling effect on the climate[5].

**Cheap electricity and high fossil fuel prices improve DACCU's
competitiveness**
Given that the economic superiority of DACCU does not hold under a
$CO_2$ neutrality target, which better reflects the current level of ambi-
tion of aviation mitigation policies[69,70], and under demand-side and
contrail mitigation options, we further investigate the conditions
under which DACCU-based fuels could be economically competitive
with an emit-and-offset strategy via DACCS and even with business-as-
usual. To this end, we perform local sensitivity analyses on the most
influential parameters and perform optimization to identify the para-
meter values that minimize the cost penalty of DACCU under $CO_2$
neutrality (Table 1 and Supplementary Table 5).

Reducing or increasing the value of input variables by 70% relative
to the default (visible in Table 1) only eliminates the advantage of
DACCS over DACCU for a change in electricity cost and fossil kerosene
price (Fig. 6a). The cost penalty of DACCU relative to DACCS is already
eliminated at an electricity cost of €0.02 kWh$^{-1}$ (Fig. 6b). This threshold
is below the 2022 levelized cost of onshore wind (€0.033 kWh$^{-1}$[71]), but
not unattainable in the future through technology learning, economies
of scale, and optimal siting[72,73]. In contrast, only when powered by free
electricity, DACCU is close to competitive with the business-as-usual
(Fig. 6c, d). Reducing the levelized cost of electricity to zero can only
be achieved if the production of DACCU-based fuels is limited to
periods of excess renewable electricity production, such as sunny
summer days in grids with a high share of solar PV, which are unlikely
to meet global jet fuel demand.

Conversely, rising fossil kerosene prices prove transformative:
DACCU becomes cost-competitive with DACCS at a fossil kerosene
price of €0.88 l$^{-1}$ and with the business-as-usual scenario at €1.27 l$^{-1}$.

## CO₂ neutrality – DACCU vs. DACCS

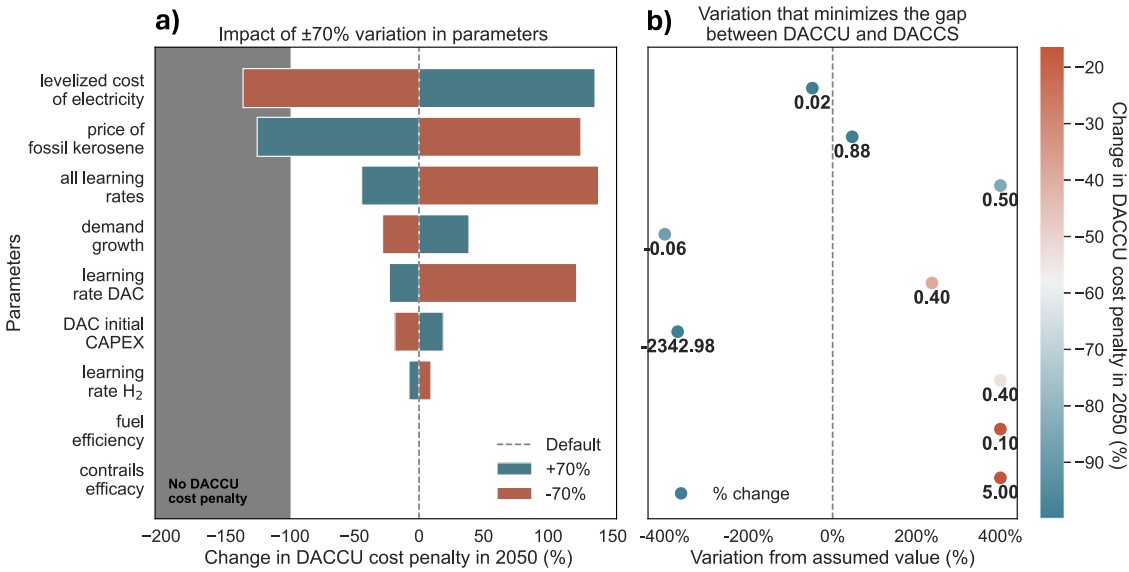

## CO₂ neutrality – DACCU vs. BAU

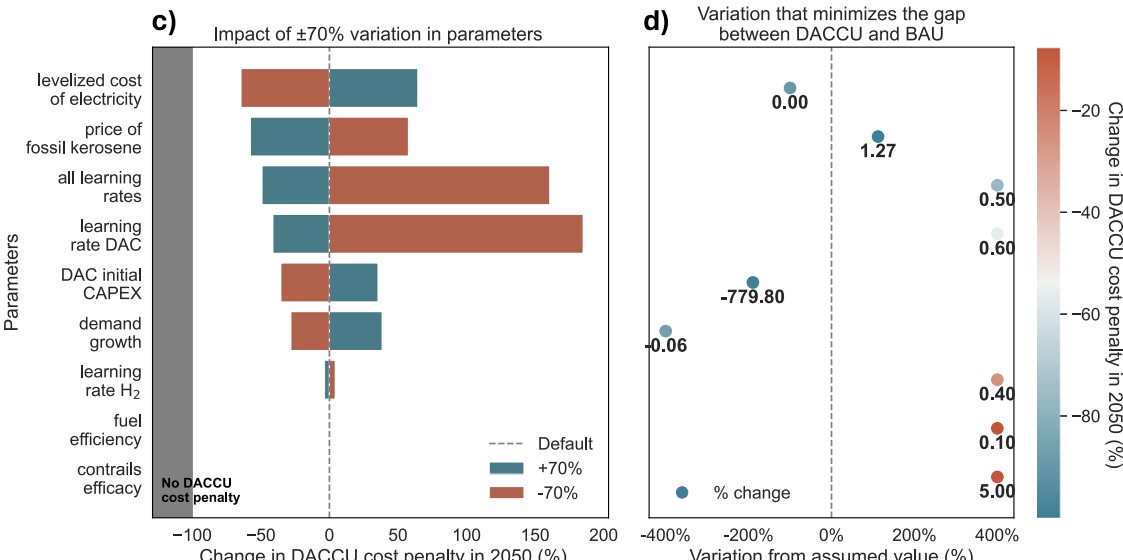

**Fig. 6 | Sensitivity analysis.** Impact of local variation in key input parameters on the Direct Air Carbon Capture and Utilization ("DACCU") cost penalty relative to Direct Air Carbon Capture and Storage ("DACCS") (**a** and **b**) and business-as-usual (**c** and **d**) to achieve CO₂ neutrality by 2050. Panels (**a**) and (**c**) describe the impact of ±70% variation in input parameters on the change in DACCU cost penalty by 2050 (%). The shaded gray area indicates where the DACCU cost penalty is eliminated i.e., DACCU becomes cost advantageous compared to either DACCS or business as usual. Panels (**b**) and (**d**) show the values at which the cost penalty of DACCU is minimized, i.e. where the difference in total cost between DACCU and either DACCS or business as usual is closest to zero. The points on the x-axis show the percentage change from the default value, the DACCU cost penalty (colors) and the new "optimal" values (numbers).

Such high fossil kerosene costs would not only make DACCU a more economical option but would likely also reduce demand. While removing current direct and indirect subsidies on fossil fuels could raise the current price of fossil kerosene to about €0.8 l⁻¹[74], higher increases in the current price of fossil kerosene (€0.6 l⁻¹) would require dedicated political ambition[75].

All other key input variables fail to close the price gap between DACCU and DACCS or the business-as-usual scenarios. Declining demand at a 6% yearly rate can reduce the gap by over 90% but does not close it. Accelerated technological learning and steeper learning curves benefit both DACCU and DACCS scenarios. Thus, even a learning rate of 50% - higher than has been observed historically for fast-learning technologies such as solar PV - cannot close the gap between the DACCU and DACCS pathways. Similarly, only negative initial DAC CAPEX values (equivalent to being paid to deploy DAC) can close the gap between DACCU and business-as-usual - a condition that is unlikely to ever materialize.

In summary, optimistic changes in fossil kerosene price or in the levelized cost of electricity are required to make DACCU cost-competitive with DACCS or business-as-usual by varying a single parameter. However, a synergy of lower electricity costs with rising fossil kerosene costs could accelerate a scenario where DACCU

outperforms DACCS or even fossil jet fuels under optimistic but possible conditions (Supplementary Fig. 11). These conditions could be created by strategic policy interventions, such as limiting DACCU-based synthetic fuel production to optimal locations or periods of significantly cheaper surplus electricity and raising the price of fossil kerosene by removing subsidies or pricing emissions (see Supplementary Fig. 12 for an exploratory analysis of the effects of such policies).

## Discussion

In this study, we investigate the conditions under which aviation mitigation via DACCU-based synthetic fuels becomes cost-competitive with an emit-and-offset strategy via DACCS. We found that these conditions are realized by either ambitious climate targets for the aviation sector that consider the non-$CO_2$ impacts of aviation, or strategic interventions that increase the price of fossil kerosene or limit DACCU to the use of cheap electricity. Additionally, we show that DACCU competitiveness diminishes if contrails are mitigated otherwise, for example via reductions in demand or rerouting flights. Finally, our analysis highlights that achieving $CO_2$ neutrality through DACCU increases flight ticket prices only slightly relative to the DACCS pathway and even relative to a business-as-usual pathway. This small price difference for consumers sheds light on the attractiveness of DACCU, which has a lower cost per avoided emissions and is consistent with broader societal goals of climate mitigation and fossil fuels phase-out.

These findings mark a departure from previous studies[22,26], which favored DACCS due to conservative assumptions about future levelized cost of electricity (which exceeds that of current wind and solar PV) and carbon-intensive energy mixes, resulting in higher lifecycle emissions of DACCU. Furthermore, due to their regional focus on Europe, where land availability is scarce, Sacchi et al. concluded that the land use of the energy-intensive DACCU pathway is a bottleneck under a scenario of continued demand growth for the aviation sector. While their regional land availability constraint does not apply to our global analysis, spatial considerations may indeed affect the cost at which the DACCU pathway could be realized due to the spatial distribution of electricity costs and the potential need for additional transportation infrastructure from remote locations.

While not inherently impractical, the ticket price surcharge due to DAC-based mitigation represents a significant increase over that imposed by current environmental policies, particularly in the short term. In 2024, Lufthansa introduced an environmental fee ranging from €3 (London-Berlin) to €72 per flight per passenger to help fund the EU mandate of 2% sustainable aviation fuel by 2025[76]. Using DAC-based solutions alone, this fee would need to increase annually by 4-8% to achieve $CO_2$ neutrality and by 6-10% to achieve climate neutrality by 2050. In the short term (up to 2030), while the relevant technologies are still in the early stages of their learning curves[77,78], the gap between current charges and the additional costs required for a transition to 100% DAC-fueled aviation is substantially larger. This would require annual fee increases of 20-32% for $CO_2$ neutrality and 22-36% for climate neutrality (Supplementary Fig. 5). This also reflects the fact that the share of DACCU synthetic fuels by 2030 in our scenario (3.7%) is substantially higher than what is mandated by the EU (1.2%)[70]. Realistically, however, achieving these goals will require a mix of solutions, including biofuels (not evaluated in this study), other CDR methods, and contrail mitigation, such as rerouting. These alternatives are likely to reduce the overall cost of aviation mitigation, making it more consistent with current efforts.

Deploying a broader range of solutions would also reduce the effort required to achieve $CO_2$-neutral and, in particular, climate-neutral flying based solely on DAC, which may not be feasible. In fact, more than 2 $GtCO_2$ of DAC would need to be installed by 2050 to achieve $CO_2$ neutrality, rising to 7 $GtCO_2$ if the goal is to offset fossil jet fuel emissions to achieve climate neutrality. These amounts of DAC far

exceed the projections of novel CDR methods by 2050 in Integrated Assessment Models simulations consistent with <2 °C targets[31,43,79], but may be possible if DAC grows at a rate comparable to some historical analogues, such as wind power[80]. However, the assumed growth rate up to 2050 (roughly 50 to 60% annually) is in line with that assumed by Integrated Assessment Models for the years between 2040-2080[31] and with that observed historically for solar PV[81]. On the other hand, by 2050, the DACCU pathway will require over 15 PWh of electricity to produce the amount of synthetic fuels necessary to fully meet global aviation demand if demand continues to grow. Given that in 2021 the global renewable energy produced amounted to 8 PWh[82], this energy demand would require a massive scale-up of renewable energy. However, DACCU's renewable energy requirements are compatible with estimates of the total technical renewable energy potential (170-270 PWh[83]).

Finally, our framework does not dynamically link the additional costs of mitigation to the likely reductions in demand. This is also justified by the fact that different policy options could shift the additional costs in a way that makes them less visible to consumers and avoids dramatic reductions in demand and thus losses in consumer surplus. Moreover, the superiority of DACCU in our results also hinges on uncertain variables, particularly the effectiveness of DACCU-based synthetic fuels in mitigating non-$CO_2$ impacts. While early empirical evidence is consistent with this trend[46,47,84], the limited number of studies evaluating the impacts of synthetic fuels, coupled with the inherent uncertainty surrounding aviation's non-$CO_2$ effects, introduces a degree of uncertainty. Notably, our analysis explicitly accounts for these uncertainties by examining scenarios in which the effects of contrails are either halved ("Rerouting") or eliminated ("Ignoring contrails").

By shedding light on the conditions that make DACCU cost-competitive, our analysis can guide policymakers in designing strategies to facilitate the competitiveness of DACCU with both a emit-and-offset pathway relying on DACCS and a business-as-usual scenario. These strategic policy interventions could be justified based on the drawbacks of the DACCS pathway associated with its reliance on fossil jet fuels and the climate mitigation benefits of DACCU fuels.

## Methods

In this study, we combined techno-economic modelling with life cycle assessment to compare the costs of mitigating the aviation sector by either compensating aviation emissions with DACCS or by replacing the whole volume of jet fuel with DACCU-based synthetic fuels, as shown in Fig. 7.

### Demand and fuel scenarios

All scenarios are based on the same demand for jet fuel, which is derived from a combination of historical data[5] from 1990 to 2018 with estimates of future demand until 2060. These are based on the assumptions of full recovery to pre-covid levels by 2024-2025 and on a 2% growth from 2024 to 2060, which are consistent with projections from various studies[1,8,10,85–88]. In addition to the total fuel demand, we also project the total annual distance flown by applying a 2% increase in efficiency, consistent with the International Civil Aviation Organization's target[69], to the historical relationship between distance flown and the amount of fuel burned[5]. While this relationship may change in the future due to an increase in long-haul flights[9,89] that burn more fuel per kilometer[90], its effect would not significantly alter the results of our analysis, as shown in our sensitivity analysis (Fig. 6 and Supplementary Fig. 13).

As detailed in the "Scenarios and Framework" section of Results, we consider two different mitigation pathways for aviation, one based on continued reliance on fossil jet fuel and offsetting through DACCS, and the other based on the gradual substitution of fossil kerosene with DACCU-based synthetic fuels. Although the American Society for Testing Material D7566 standard[91] currently allows only up to 50% synthetic fuels blends, we assume that aircraft will operate on 100%

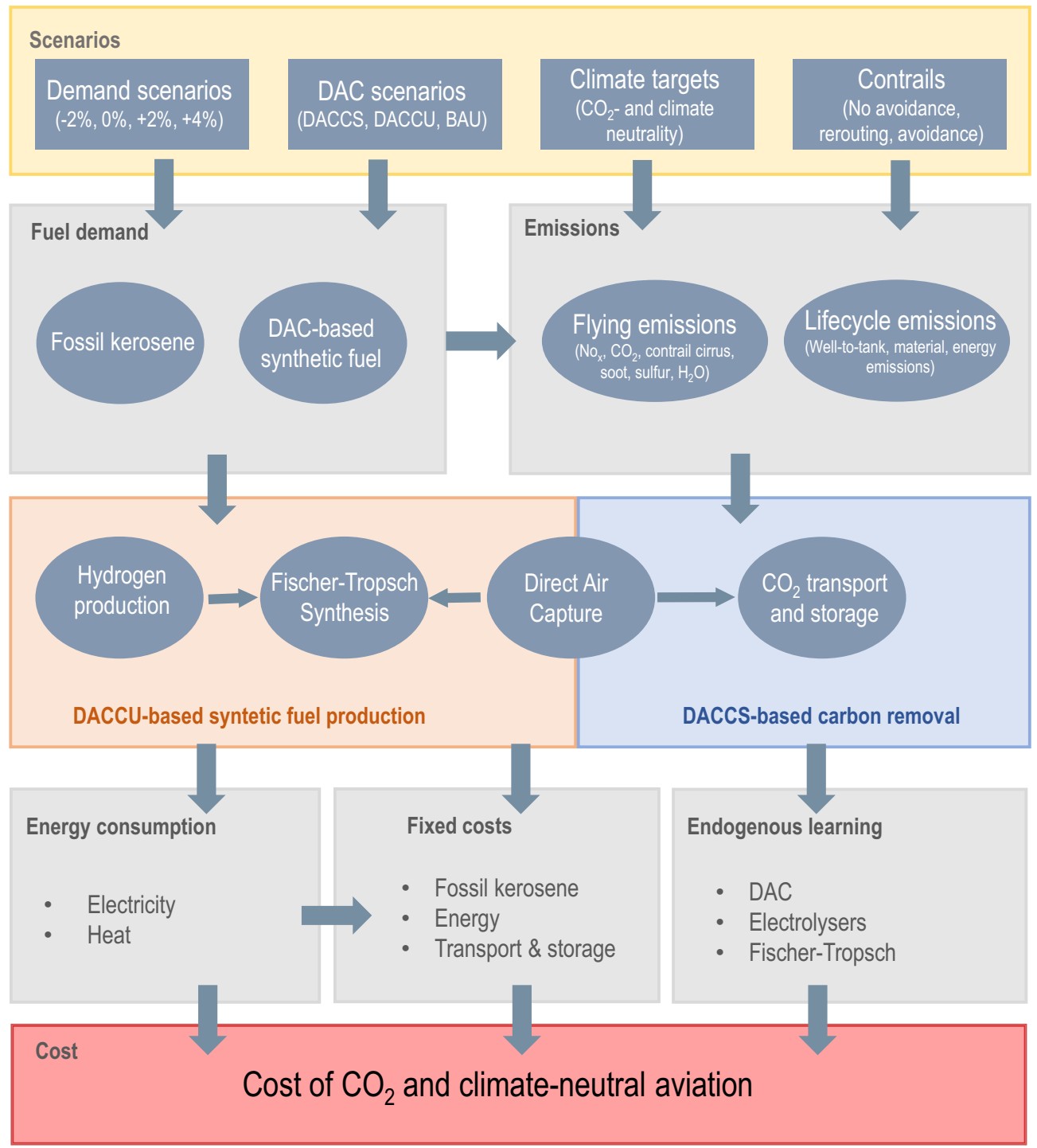

**Fig. 7 | Overview of modelling framework.** Input scenarios are in yellow, in-between calculations in grey, with orange and blue boxes to indicate the different technologies belonging to either the Direct Air Capture and Utilization ("DACCU") or Direct Air Capture and Storage ("DACCS") pathways. In red we show the final output, namely the cost of each pathway.

DACCU-based synthetic fuels by 2050, being expected that blends up to 100% will be certified in due course, so that planes can fully run on synthetic fuel. Similarly, we model an upscaling of DACCS that enables full offsetting of aviation emissions by 2050, simplistically assuming no constraints on the rate of adoption of this technology.

**Emissions and offsets**

To calculate the amount of direct emissions from fossil jet fuel combustion, we apply the relationships between fossil jet fuel and $CO_2$, water vapor, sulfur dioxide, soot, and $NO_x$ emissions reported by Lee et al.[5]. Contrail cirrus formation was calculated using the relationship between the distance flown contrail length, also reported in Lee et al.[5]. To calculate the emissions and contrail clouds formation of DACCU-based fuels, we follow the approach described in Brazzola et al.[25] (see their Supplementary Table 2) and propagate their uncertainty ranges throughout the analysis.

The direct flight emissions then drive the demand for carbon removals via DACCS to offset their climate impact. The amount of

removals is further determined by (1) the specific climate target chosen (i.e., $CO_2$ or climate neutrality, see Figs. 1)), and (2) the lifecycle emissions of each technology pathway. First, to achieve $CO_2$ neutrality, we simplistically assume that we can fully compensate the climate impact of one ton of $CO_2$ by removing an equivalent amount via DACCS, neglecting the uncertainties of this relationship[92,93]. To achieve climate neutrality, we compensate for the non-$CO_2$ effects with DACCS based on the GWP* metric following the approach of Brazzola et al.[25] and using their 'Gold' definition of climate neutrality[25]. Thereby, we use and propagate throughout the analysis the uncertainties in the relationship between non-$CO_2$ emissions and their effective radiative forcing reported in Lee et al.[5].

Finally, we also offset through DACCS the difference in lifecycle emissions due to the material and energy footprint of the two pathways, so that they both achieve the same level of emissions. We calculate lifecycle emissions for both fossil kerosene, DACCU-based fuels, and DACCS. For fossil kerosene, we considered the well-to-tank emissions from Moretti et al.[94], which reflect European averages. Future reductions in oil refining emissions are based on the oil industry decarbonization prospects[95], leading to a progressive decrease in well-to-tank emissions for fossil jet fuels. Material footprints are based on values for the production of required adsorbents and DAC modules by Deutz and Bardow[96]; values for electrolysers by Delpierre et al.[97]; values for CO electrolysis production units from Adnad and Kibria[98]. In addition, we calculated the energy requirements of all technologies involved and applied an electricity carbon footprint for an average global electricity grid[96], assuming high decarbonization efforts over time leading to near net-zero emissions in 2060. As the synthesis of DACCU-based fuels is a multi-functional unit process with by-products, notably diesel, we assume the production of 0.82 tons of diesel per ton of jet fuel[22]. The lifecycle inventory of the unit processes up to the Fischer-Tropsch unit was then allocated to jet fuel by means of mass allocation (resulting in a 54.5% share for jet fuel).

### Techno-economic assessment of DACCS and DACCU pathways

Finally, we calculate the energy consumption and capital costs of each technology and fuel included in the DACCS and DACCU pathways from 2020 to 2060 (Supplementary Fig. 4). This includes the cost of fossil jet fuel, electricity and heat consumption, $CO_2$ transport and storage, and the capital costs of DAC, $CO_2$ reduction, electrolysis, and Fischer-Tropsch synthesis.

For both pathways, we consider a low-heat solid-sorbent DAC system. While high-temperature liquid-solvent DAC may be more energy-efficient for the production of DACCU-based fuels, there are currently no plants that operate completely without burning natural gas[99]. As a result, using liquid-solvent DAC to produce jet fuel may result in net $CO_2$ emissions. We moreover assume a fixed cost of 20 € $tCO_2^{-1}$ for $CO_2$ transport and storage as in Becattini et al.[22], based on the assumption that DACCS would be optimally located next to storage sites.

For the production of DACCU-based synthetic fuels, we introduce some variance by considering four different combinations of two water electrolysers (either polymer membrane or alkaline electrolysers) and two $CO_2$ reduction methods (electrochemical $CO_2$ reduction and reverse-water-gas-shift). While we also calculate total costs for each technology configuration (Supplementary Fig. 6), in the main results we use an average of the costs of all four possible configurations since we cannot predict which technology will ultimately prevail due to the low technological maturity, uncertain future development, and trade-offs in terms of cost and energy intensity of different technologies involved in DACCU-based synthetic fuel production.

We first derive the installed capacities of each technology from the amounts of synthetic jet fuel required and from calculations of DACCS-based offset, as explained in the previous sections. To calculate their costs and energy consumption, we apply the parameters and assumptions summarized in Table 1 and Supplementary Tables 1 and 5.

To calculate changes in energy efficiency, we polynomially interpolate between current values and future estimates (Supplementary Table 3). In the case of CAPEX, we apply a learning rate following Eq. (1):

$$CAPEX(t) = CAPEX(t_0) * \left(\frac{Q_t}{Q_{t_0}}\right)^{-b} \tag{1}$$

Where $Q$ is the quantity of installed capacity of a technology and $b$ equals $\log_2(1 - LR)$, and $LR$ is the learning rate.

To calculate the increase in ticket price per passenger for three representative flights (London-Berlin, London-New York, London-Perth), we first calculate the cost of achieving $CO_2$ or carbon neutrality per kilometer flown each year. We then calculate the non-fuel operating expenses based on the share of kerosene in total operating costs for different routes as reported to Ringbeck et al.[100]. In the business-as-usual scenario, we calculate future ticket prices by combining the cost for fossil kerosene for the three routes, which reflects efficiency gains, with non-fuel operating costs, which we assume to remain constant. To this, we apply a 25% mark-up, reflecting the gross profit margin according to U.S. airline data[101], which corresponds to a 3% net margin after tax. Finally, to assess the impact on ticket prices of deploying DACCS and DACCU, we replaced the kerosene costs in the business-as-usual scenario with the costs required to achieve $CO_2$ or climate-neutral aviation based on either DAC approach. The relevant parameters for these calculations are shown in Supplementary Table 4.

Finally, we conduct a local sensitivity analysis on key parameters highlighted in Table 1 as well as in Supplementary Table 1, systematically varying uncertain input parameters by fixed percentages to ensure comparability (Supplementary Figs. 12–14). We additionally also explore the effect of different demand scenarios and contrail mitigation scenarios by varying key input parameters (Supplementary Table 5, Supplementary Fig. 7–9)

## Data availability

All data generated in this study are available in a GitHub repository under accession code https://github.com/nikibraz/DACCSvsDACCUaviation.git (https://doi.org/10.5281/zenodo.14185744).

## Code availability

The code to reproduce this analysis is available at https://github.com/nikibraz/DACCSvsDACCUaviation.git (https://doi.org/10.5281/zenodo.14185744).

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

## Acknowledgements
N.B. is funded through a Doc.CH Grant No. P000PS_203884 and acknowledges support from the Swiss National Academy of Science. C.M.'s work on this study was sponsored by the Swiss Federal Office of Energy's "SWEET" programme as part of the reFuel.ch project under Grant Number SI/502717.

## Author contributions
N.B., A.P., and T.T. conceived the idea of this study. N.B., A.M., and C.M. designed the study. N.B. and A.M. performed the analysis with support from C.M. and T.T. N.B. wrote the manuscript. N.B., T.T., A.P., and C.M. edited the manuscript and engaged in ongoing discussions.

## Competing interests
The authors declare no competing interests.
