## [Peer Review File · Nature Communications]

REVIEWER COMMENTS

Reviewer #1 (Remarks to the Author):

NCOMMS-24-15241-T: Synthetic fuels may be a cheaper way to achieve climate-neutral aviation

The manuscript analyzes the role of direct air capture for mitigating aviation, via DACCS or via DACCU. The authors analyze costs, life-cycle emissions, energy consumption and technology readiness for both DAC options and find that DACCU can more effectively mitigate aviation if non-CO₂ climate impacts are considered, and if electricity prices are low and fossil fuels are expensive. This is a very relevant topic and the manuscript is insightful and well written. I have only small comments and one important suggestion for the authors, which would improve their manuscript.

Major comments

- A major concern relates with the assumed demand for. Demand is expected to grow into the future, and demand in 2050 could be at many different levels. The authors have done a very thorough analysis, but I think that they should do sensitivity cases where they analyze their results with different demand projections (other than their 2% growth), especially since industrial projections (IATA and ICAO) have scenarios with faster growth rates, and even 1980-2019 demand growth has been about 4%. This would be very helpful in understanding the costs. How feasible would it be to meet aviation's demand and keep climate goals? How much more would it cost? What are the sensible parameters here? I think these are all important questions, especially with such an uncertain parameter (future aviation demand).

- An important point to note is that likely aviation's mitigation will be done via a large suit of technologies, including some offsets, some synthetic fuels, probably some biofuels, and some share of hydrogen and electric planes. The authors should touch on this in their discussion. Although assuming the "either-or" approach for their paper is a useful thought experiment, it is likely that in reality this does not happen, and the authors should at least acknowledge this facet.

Minor comments

- I like how the authors title their sections with the main finding. For example: "Do non-CO₂ emissions decrease with synthetic fuels?"

- The authors should discuss why they decided to analyze 2050. Most likely because of emission targets. It would be good to tie this to actual policies.

- Figure 1

o Overall: it is a great figure!

o However, it is a little confusing that in the “emit and offset” scenario there are offsets, and that in the synthetic fuel scenario there are also offsets. Unless I misunderstood the paragraph before, the offsets in the “emit-and-offset” scenario are directly from DAC, while the “offsets” from the “synthetic fuels” scenario in reality come from fuel switching, not necessarily from DAC. Is this correct? If so, maybe the figure needs to be relabeled as “carbon reductions” instead of carbon removals/offsets, because it includes both the reduction from fuel switching, and the reduction from actually using DAC. I just find this part of the figure a little confusing. Another option would be to make a waterfall plot showing the three scenarios, so that the actual reduction in emissions until reaching 0 is more visual.

- I suggest the authors call out the scenarios in the text. Currently the text says there are two scenarios the “DACCU scenario” and the “DACCS scenario”. It would be useful to put in parenthesis the names used in Figure 1 DACCU (synthetic fuels scenario) and DACCS (emit-and-offset) scenario. It would be easier for the reader to understand.

- Line 103 “assuming rising aviation demand”. I would specify by how much. And note my comment on the “major comments” section.

- I would add in the caption of Figure 2 what do the black brackets represent. Is it an uncertainty parameter? If so, based on what?

- Line 196 “This threshold is well below the 2023 price of the cheapest renewable energy sources, onshore wind”, which costs how much?

- Authors should discuss why, if costs of plane tickets would only modestly increase (20-55 Euros for a long-haul flight), why we are not doing this now? What are the major hurdles today? Which specific policies could help overcome them? The authors talk about the different sensible parameters (such as electricity prices), but they can go a little further and suggest actual policies.

- The column names on Figure 5 could be a little more clear: there is “learning rate”, “learning rate H2”, and “learning rate DAC”. What is “Learning rate” alone? Growth rate of what? Fuel efficiency of what?

- Sentence on Row 243 is a run-on sentence, and it is not clear: “In contrast to direct subsidies...”. Authors should rephrase it.

Reviewer #2 (Remarks to the Author):

Dear authors and editor.

It was a pleasure to go through your manuscript. It is a relevant topic and well explained. To improve further, consider my comments.

1)

Figure 1:

This indicates that the non-CO2 effects are larger than the CO2-effects in aviation. Is this really the case? On other words: is the figure only explaining the concept or does it also indicates some real world ratio's? I would opt for the latter, but I have no clue about the division.

In that respect: also give an indication/percentage in line 88-90 to what extent they burn cleaner.

2) line 116-118

a)

Is the 500 billion cumulative towards 2050? Or is it the higher cost in 2050? Figure 2 states Euro/year so points the latter, where I am more interested in the cumulative cost difference.

Please make clear so that it is easy to understand without coming back to it after reading the report.

b)

please briefly explain the business as usual scenario here. Particularly in terms of carbon pricing and exiting fossil fuel subsidies. Is it the current setting globally or do you include a pathway of more carbon pricing and less subsidies?

3) Figure 2

The 200 and 500 billion numbers that are referred to in the text are not intuitively easy to read from the graph. I advise to present the y-axis in billions so it runs from 0 to 2000, right?

4) line 149

I would end that line with something like:

....., but here the cost difference is much smaller compared to the CO₂-neutrality target.

5) line 158-162

I struggle to grasp whether this is from a societal or investors perspective. As a financial engineer at a bank I tend to focus on capex/unit first. They get lower as DAC installations grow at the same pace in both scenario's right? So an investor has the same capex/unit in both scenarios for the same installed capacity. So I presume you are talking societal costs here, right? Please make clear with some extra words.

6) line 195-196

There are so many power prices. I guess you are talking wholesale power prices on exchanges here (without taxes and transportation costs), or end prices for the operator of the electrolyser (including transport/grid costs and taxes?).

Currently this makes a lot of difference for the operator of an electrolyser and actually prevents the take up of green hydrogen in Europe right now, particularly in the North-Western European hydrogen market.

Also, it is more common practice to state power prices in euro/MWh for heavy power users.

7) line 202-206

I would expect a reference to the large implicit fossil fuel subsidies that aviation gets. Taxing is more a matter of removing those implicit subsidies that are in place for so long compared to other sectors and smaller users.

8) line 243-253

As an economist I don't buy the assumption that electrolyzers will only run in limited hours where power prices are low. They are so capital intensive that they need to run almost continuously to recoup investment costs. Working for a bank I know bankers are not keen on financing electrolyzers that stand idle for most of the time.

9) Discussion section

Overall feeling after reading the article:

You focus on DAC and CCUS technologies only and you state that in the introduction. For example by stating some of the problems with biomass. Still, DAC is also not without problems/controversy and highly debated. I wonder how DAC economically compares to other Carbon Removal Technologies like biomass, algae, limestone, etc.

Would be great if you can add a paragraph on that as other sources point to the fact that DAC technologies are relative costly ways of carbon removal.

10) References

It would be nice to reference my article on SAF:

<https://think.ing.com/articles/synthetic-fuels-answer-to-aviations-net-zero-goal>

Although I do understand that it is a 'less scientific' source.

Success with finalising the article!

Happy to get a digital copy once it is published.

Reviewer #3 (Remarks to the Author):

The basic message of the manuscript, that is, that--under certain conditions--synthetic fuels from DAC and hydrogen can result in lower mitigation costs compared to a pathway with continued reliance on fossil-based jet fuel with DAC and carbon storage is plausible. Yet, I have got a couple of concerns, some of which can be addressed relatively easily, others would require more work.

Many of the numbers presented in the manuscript depend heavily on assumptions, which are not presented in the main body of the text. For example, on page 3, the authors state that the DACCS pathway experiences lower costs "by about €200 billion in 2050 (Figure 2) and €120 billion in 2060". However, the underlying oil or jet fuel price, electricity prices, technology cost assumptions, etc., are reported only in SI Table 1. I strongly believe that the key assumptions, particularly the oil and electricity prices, MUST be presented early on, otherwise the results cannot be interpreted. Only SI Table 1 seems to report "Cost of fossil fuels" as Euros 0.6/L. Presumably these are the jet fuel prices? Why this number, what is the rationale?

The critical role of the jet fuel price in that regard can be seen also from Figure 2. The y-axis of this figure is labelled as "costs", whereas the caption states "Cost to achieve CO₂ and climate neutrality..." In order to be more meaningful, the authors should subtract the costs of fossil jet fuel use in the baseline scenario in order to illustrate the EXTRA or MARGINAL costs for achieving these targets. Also, shouldn't this figure show cumulative discounted costs instead of year 2050 costs? The magnitude of the numbers of the y-axis suggests this, but it is not clear from the text. If yes, what is the discount rate?

As partly covered in the above comments, the authors need to distinguish clearly between costs and prices. In the current manuscript, these terms seem to be used interchangeably, particularly wrt "fossil fuels" (do the authors mean "jet fuel"?) and electricity.

I also suggest simplifying the multiple coloured tables in the text. Which of these variables matter most? Can this be shown graphically?

The authors also state that both pathways penetrate the fuels market following S-curves. Can this be shown graphically? I have difficulties imagining the feasibility of adding thousands of large-scale DAC systems and potentially synthetic fuel plants within only 25 years if following an S-curve.

My main concern relates to the overarching framework. Presumably as a way to ensure consistency, "All scenarios are based on the same demand for jet fuel" (Methods section). However,

the different fuel costs associated with achieving CO2 or climate neutrality following the two pathways will result in different airfares and thus reductions in air transportation demand. This, in turn, will mitigate the demand for fuels along with CO2 emissions/climate impact, but also result in different losses of consumer welfare. According to my point of view, a rigorous analysis needs to ensure consistency based on fundamental economics. Adding this step would require additional work, but significantly strengthen the economic foundation of the analysis.

Can synthetic fuels achieve climate-neutral aviation in a cost-effective way?

We would like to thank the reviewers and the editor for their helpful and positive feedback. We believe that your comments have enabled us to substantially improve the manuscript.

In this document, we provide **answers** to all **comments** made by the reviewers. We provide our **answers in blue**. Moreover, we highlighted in the manuscript attached the sentences and paragraphs that have been substantially modified to address the concerns of the editor or of more than one reviewer in **red**, of reviewer 1 in **green**, of reviewer 2 in **brown**, and of reviewer 3 in **purple**.

REVIEWER COMMENTS

Reviewer #1:

The manuscript analyzes the role of direct air capture for mitigating aviation, via DACCS or via DACCU. The authors analyze costs, life-cycle emissions, energy consumption and technology readiness for both DAC options and find that DACCU can more effectively mitigate aviation if non-CO₂ climate impacts are considered, and if electricity prices are low and fossil fuels are expensive. This is a very relevant topic and the manuscript is insightful and well written. I have only small comments and one important suggestion for the authors, which would improve their manuscript.

Minor comments

- I like how the authors title their sections with the main finding. For example: "Do non-CO₂ emissions decrease with synthetic fuels?"

Thank you for the positive review and helpful suggestions, which we have implemented and which we believe have contributed to a significant improvement of the manuscript. In the text, you can see the changes that result specifically from your comments in **green**, and the changes that result from a mix of your comments and those of other reviewers and the editor in **red**.

Comment #1

A major concern relates with the assumed demand for. Demand is expected to grow into the future, and demand in 2050 could be at many different levels. The authors have done a very thorough analysis, but I think that they should do sensitivity cases where they analyze their results with different demand projections (other than their 2% growth), especially since industrial projections (IATA and ICAO) have scenarios with faster growth rates, and even 1980-2019 demand growth has been about 4%. This would be very helpful in understanding the costs. How feasible would it be to meet aviation's demand and keep climate goals? How much more would it cost? What are the sensible parameters here? I think these are all important questions, especially with such an uncertain parameter (future aviation demand).

Response #1

Thank you very much for your valuable comment. We did indeed include a sensitivity analysis of different demand growth rates in our original submission. The parameter was labeled "growth rate" which could have been misleading. We have now clarified this in the revised sensitivity analysis figure (Figure 6) to make it explicitly clear that it refers to demand.

In addition, your comment has prompted us to undertake a more comprehensive analysis of different demand scenarios, as demand reduction is in itself a possible approach to mitigate the climate impact of aviation. We have added a new section in the Results (lines 196-235) with one additional Figure (Figure 5) illustrating this analysis. We have also included Supplementary Figures 7-9 for a more detailed comparison of the impact of different demand scenarios. These scenarios consider both faster growth rates and potential reductions in demand (2% increase, historical growth at 4%, flat demand, and a 2% reduction in demand). These additional figures are also referenced in the main text (lines 220-229), where we highlight the challenges of achieving 2% demand growth, particularly given the historical growth rate of 4%. Our results indicate that DACCU loses its cost-competitiveness under climate neutrality if demand is reduced or if contrail formation is avoided. As a result, we have revised the Abstract and Discussion to reflect these findings (lines 16-22, lines 288-289).

Comment #2

An important point to note is that likely aviation's mitigation will be done via a large suite of technologies, including some offsets, some synthetic fuels, probably some biofuels, and some share of hydrogen and electric planes. The authors should touch on this in their discussion. Although assuming the "either-or" approach for their paper is a useful thought experiment, it is likely that in reality this does not happen, and the authors should at least acknowledge this facet.

Response #2

We completely agree with your observation. We have now included this point in the discussion (lines 317-324) and in the "Scenarios and Framework" section of the results (lines 77-78). In addition, your comment has prompted us to remark that there may be more effective strategies for reducing non-CO₂ emissions and contrails than relying solely on synthetic fuels and DACCS. For example, alternatives such as flight rerouting or the reduction in the aromatics content of fossil kerosene, via hydrogenation, may be more effective. As a result, we have expanded our previous analysis to include an entire section on the implications of managing contrails differently (lines 196-235), and updated the Supplementary Information (cf. Supplementary Figures 9). We have also expanded the motivation of our sensitivity analysis taking into account this comment (lines 238-242).

Comment #3

The authors should discuss why they decided to analyze 2050. Most likely because of emission targets. It would be good to tie this to actual policies.

Response #3

Thank you for this important point. We agree that tying our analysis to actual policies is crucial. We chose the year 2050 primarily due to its significance in national emission targets and added this justification in lines 79-81. However, the exact date of achieving net-zero emissions does not significantly impact our results because of factors such as endogenous

learning, the unconstrained scale-up of technologies, and the absence of discounting in our model. The only aspect that the date influences is the cost difference between DACCS and DACCU for achieving CO₂ neutrality, which shrinks when reaching net-zero emissions later, such as by 2060. The later net-zero emissions are reached, the higher are emissions by the time net-zero is achieved, and thus the more emissions need to be counterbalanced via CDR in the DACCS pathway. To show this effect, we have added Supplementary Figure 10 illustrating a scenario where net-zero emissions are reached by 2060 and discuss this scenario in the manuscript at lines 81-82 and 123-125.

Comment #4

Figure 1 - Overall: it is a great figure!

However, it is a little confusing that in the “emit and offset” scenario there are offsets, and that in the synthetic fuel scenario there are also offsets. Unless I misunderstood the paragraph before, the offsets in the “emit-and-offset” scenario are directly from DAC, while the “offsets” from the “synthetic fuels” scenario in reality come from fuel switching, not necessarily from DAC. Is this correct? If so, maybe the figure needs to be relabeled as “carbon reductions” instead of carbon removals/offsets, because it includes both the reduction from fuel switching, and the reduction from actually using DAC. I just find this part of the figure a little confusing. Another option would be to make a waterfall plot showing the three scenarios, so that the actual reduction in emissions until reaching 0 is more visual.

Response #4

Thank you for the positive feedback on Figure 1. We understand that there is some confusion about the sources of offsets in the different scenarios. To clarify, the offsets in the “emit-and-offset” scenario (now called simply ‘DACCS’ to address Comment #5 of this Reviewer) are indeed directly from DACCS, whereas in the “synthetic fuel” scenario (now called simply ‘DACCU’) we have two cases: under CO₂ neutrality, the “reductions” in emissions are only due to fuel switching; under climate neutrality, in addition to these emission reductions, we also use CDR (DACCS) to offset the remaining non-CO₂ emissions.

To make this distinction clear, we have revised Figure 1 to show the actual breakdown of emissions into different components for both the Business-as-Usual, CO₂ neutrality, and climate neutrality. In addition, we have updated the labels to “CDR” instead of “offsets” to better reflect the sources of emission reductions. We have also expanded the Figure caption for better understanding (l. 753-762).

Comment #5

Figure 1 - I suggest the authors call out the scenarios in the text. Currently the text says there are two scenarios the “DACCU scenario” and the “DACCS scenario”. It would be useful to put in parenthesis the names used in Figure 1 DACCU (synthetic fuels scenario) and DACCS (emit-and-offset) scenario. It would be easier for the reader to understand.

Response #5

We agree with the need for consistent naming and have therefore revised the text to include the names used in Figure 1, namely DACCS and DACCU, together with new text recalling their meaning directly in the caption of the figure (l. 753-762).

Comment #6

Authors should discuss why, if costs of plane tickets would only modestly increase (20-55 Euros for a long-haul flight), why we are not doing this now? What are the major hurdles

today? Which specific policies could help overcome them? The authors talk about the different sensible parameters (such as electricity prices), but they can go a little further and suggest actual policies.

Response #6

We have added an additional paragraph in the discussion (lines 305-321) to address this point. In this section, we contextualize the modest ticket price increases in the context of EU policies and show that achieving the price increases induced by DAC-based mitigation would require a significant increase in ambition, leading to a fast uptake of commercial production of DAC-based synthetic fuels. We also discuss the short-term challenges, in particular showing that we need to substantially increase ambition in the short term. This expanded discussion includes an implicit policy recommendation by highlighting the higher level of ambition of our synthetic fuel shares by 2030 compared to ReFuelEU. In addition, we refer to a new Supplementary Figure that illustrates the short-term (2030) costs on a pathway to CO₂ and climate neutrality (Supplementary Figure 5).

Minor comments

- Line 103 “assuming rising aviation demand”. I would specify by how much. And note my comment on the “major comments” section.
- I would add in the caption of Figure 2 what do the black brackets represent. Is it an uncertainty parameter? If so, based on what?
- Line 196 “This threshold is well below the 2023 price of the cheapest renewable energy sources, onshore wind”, which costs how much?
- The column names on Figure 5 could be a little more clear: there is “learning rate”, “learning rate H2”, and “learning rate DAC”. What is “Learning rate” alone? Growth rate of what? Fuel efficiency of what?
- Sentence on Row 243 is a run-on sentence, and it is not clear: “In contrast to direct subsidies...”. Authors should rephrase it.

Response

Thanks for this, we have implemented all suggested changes. These are reflected in lines 91-93, 763-766, 249-250, and Figure 6.

Reviewer #2

Dear authors and editor.

It was a pleasure to go through your manuscript. It is a relevant topic and well explained. To improve further, consider my comments.

We greatly appreciate your positive review and valuable suggestions. We have incorporated your suggestions, which have significantly improved the manuscript. In the text, changes that were made specifically in response to your comments are highlighted in **brown**, while changes that resulted from a combination of your comments, feedback from other reviewers, and the editor's input are highlighted in **red**.

Comment #1

Figure 1: This indicates that the non-CO₂ effects are larger than the CO₂-effects in aviation. Is this really the case? On other words: is the figure only explaining the concept or does it also indicates some real world ratio's? I would opt for the latter, but I have no clue about the division. In that respect: also give an indication/percentage in line 88-90 to what extent they burn cleaner.

Response #1

While the Figure in the previous manuscript version was simply a schematic to explain the concept, it was indeed intended to illustrate real-world relationships and show that non-CO₂ effects can be more significant than CO₂ effects in aviation. To clarify this, we have now modified Figure 1 so as to use real emission numbers which clearly show these relationships (l. 110-112, l. 753-762). In addition, we have provided Supplementary Figure 1, which shows the time series of both emissions and radiative forcing due to different species from both the DACCS and DACCU scenarios. We hope that these additions effectively address your concerns.

Comment #2

Line 116-118

a) Is the 500 billion cumulative towards 2050? Or is it the higher cost in 2050? Figure 2 states Euro/year so points the latter, where I am more interested in the cumulative cost difference. Please make clear so that it is easy to understand without coming back to it after reading the report.

Figure 2: The 200 and 500 billion numbers that are referred to in the text are not intuitively easy to read from the graph. I advise to present the y-axis in billions so it runs from 0 to 2000, right?

b) please briefly explain the business as usual scenario here. Particularly in terms of carbon pricing and exiting fossil fuel subsidies. Is it the current setting globally or do you include a pathway of more carbon pricing and less subsidies?

Response #2

a) You make an important point. The 500 billion figure was indeed simply the cost difference in 2050, not the cumulative cost difference up to 2050. Recognizing that, as you point out, readers may be more interested in other units, we have updated Figure 2 to show the cost per liter of fuel, which is more intuitively comparable to a business-as-usual case. We have updated the text to make this explicit (lines 119, 132-134). In addition, we have included both the annual cost for each year through 2050 in Supplementary Figure 4 and the cumulative cost difference in Supplementary Figure 9, which is referenced in the main text lines 134-135.

b) We appreciate your point that the business-as-usual scenario needs further clarification.

We have now expanded the "Scenarios and Framework" section to describe business-as-usual (l. 89-91), which also addresses Reviewer 3's Comment #1. We have also added the new Table 1 (l. 796-798) showing the input assumptions underlying all scenarios.

Comment #3

Line 195-196: There are so many power prices. I guess you are talking wholesale power prices on exchanges here (without taxes and transportation costs), or end prices for the operator of the electrolyser (including transport/grid costs and taxes?).

Currently this makes a lot of difference for the operator of an electrolyser and actually prevents the take up of green hydrogen in Europe right now, particularly in the North-Western European hydrogen market.

Also, it is more common practice to state power prices in euro/MWh for heavy power users.

Response #3

Thank you for your insightful comment, which was also raised by Reviewer #3 and reinforces the need for clarity in this section. To clarify, we are referring to the levelized cost of electricity, which includes various cost components such as generation, transportation, and taxes. We have taken this information from an IRENA report and kept it in the same unit that IRENA reports, which is €/kWh. Recognizing the importance of precision, especially for the electrolyzer operator, we have made sure to state that we refer to the levelized cost of electricity and not the wholesale power prices throughout the manuscript (e.g., lines 18, 247, 249, 253, 273) and in Figure 6.

Comment #4

Line 202-206: I would expect a reference to the large implicit fossil fuel subsidies that aviation gets. Taxing is more a matter of removing those implicit subsidies that are in place for so long compared to other sectors and smaller users.

Response #4

Thank you for that important observation. We agree that it may be more appropriate to frame the issue as subsidy removal rather than an emissions tax. However, current fossil fuel subsidies account for about 10% of the price of fossil fuel kerosene, and their removal alone might not significantly increase the cost of fossil fuels. We have now included a discussion of how the removal of these subsidies could contribute to an increase in fossil fuel prices in lines 260-263 and 277-281 and in the Abstract in lines 19-21.

Regarding the taxation aspect, we have now eliminated the policy section of the results and moved its key figure to the Supplementary Information (Supplementary Figure 10), recognizing that the lack of a comprehensive and dynamic model prevents us from fully capturing the complexity and dynamics of such policies, also to address Reviewer #3's Comment #6. Instead, we have improved our analysis by incorporating more detailed sensitivity analysis and optimization to identify the "breakthrough" value of input parameters where the price gap between scenarios closes (l. 242-245, 248-249, 257-260, 264-266, 269-271, and Figure 6b and 6d). This should provide a clearer and more thorough understanding of the potential impact.

Comment #5

Line 243-253 - As an economist I don't buy the assumption that electrolysers will only run in limited hours where power prices are low. They are so capital intensive that they need to run almost continuously to recoup investment costs. Working for a bank I know bankers are not keen on financing electrolysers that stand idle for most of the time.

Response #5

We agree that the assumption that electrolyzers would only operate during periods of low electricity prices is unrealistic, given their capital-intensive nature and the need for continuous operation to recoup investment costs. In light of this, and also to address Reviewer #3's comment #6, we have removed the policy scenario analysis from the Results section and instead included these considerations in the sensitivity analysis results (lines 248-250). Thereby, we have qualified the "breakthrough" values we calculated by emphasizing the unlikely of achieving such low prices (l.253-256), thereby providing a more realistic outlook.

Comment #6

9) Discussion section - Overall feeling after reading the article:

You focus on DAC and CCUS technologies only and you state that in the introduction. For example by stating some of the problems with biomass. Still, DAC is also not without problems/controversy and highly debated. I wonder how DAC economically compares to other Carbon Removal Technologies like biomass, algae, limestone, etc.

Would be great if you can add a paragraph on that as other sources point to the fact that DAC technologies are relative costly ways of carbon removal.

Response #6

You raise an important point. In the introduction, we mention scalability as a reason for pursuing DAC in aviation, and justify the fairness of funding early DAC deployment by imposing its costs on airline passengers. However, this point may be overlooked later in the paper. Therefore, we have now added multiple acknowledgments and discussions that DAC is likely to be only part of the solution and that there are less scalable but cheaper options which will play an important role (lines 77-79, 317-321, 322-324), reflecting your concern and also addressing Reviewer 1's Comment #2.

Minor comments and clarifications

Line 149 - I would end that line with something like:

....., but here the cost difference is much smaller compared to the CO₂-neutrality target.

Implemented as suggested, thanks (lines 161-162).

Line 158-162: I struggle to grasp whether this is from a societal or investors perspective. As a financial engineer at a bank I tend to focus on capex/unit first. They get lower as DAC installations grow at the same pace in both scenario's right? So an investor has the same capex/unit in both scenarios for the same installed capacity. So I presume you are talking societal costs here, right? Please make clear with some extra words.

You are right, our analysis takes into account the societal perspective. We have now clarified in the manuscript (Methods, Figure 7) that DAC costs decrease by applying endogenous learning rates, i.e., larger and faster DAC deployments lead to larger and faster decreases in DAC CAPEX. This results in different unit DAC costs in our analysis due to different total DAC volumes across scenarios (see Supplementary Figure 2). We have added this clarification (line 168) and a reference to the SI figure to make this distinction clearer.

10) References

It would be nice to reference my article on SAF:

<https://think.ing.com/articles/synthetic-fuels-answer-to-aviations-net-zero-goal>

Although I do understand that it is a 'less scientific' source.

It is an interesting publication, strictly linked topic-wise. We added a reference at line 263.

Succes with finalising the article!
Happy to get a digital copy once it is published.

Reviewer #3

The basic message of the manuscript, that is, that--under certain conditions--synthetic fuels from DAC and hydrogen can result in lower mitigation costs compared to a pathway with continued reliance on fossil-based jet fuel with DAC and carbon storage is plausible. Yet, I have got a couple of concerns, some of which can be addressed relatively easily, others would require more work.

Thank you for the insightful review. Your feedback has contributed to a significant improvement of the manuscript, and we appreciate the contribution to a better economic framework of the analysis. In the text, you can see the changes that result specifically from your comments in **purple**, while the changes that result from a mix of your comments and those of other reviewers and the editor are in **red**.

Comment #1

Many of the numbers presented in the manuscript depend heavily on assumptions, which are not presented in the main body of the text. For example, on page 3, the authors state that the DACCS pathway experiences lower costs "by about €200 billion in 2050 (Figure 2) and €120 billion in 2060". However, the underlying oil or jet fuel price, electricity prices, technology cost assumptions, etc., are reported only in SI Table 1. I strongly believe that the key assumptions, particularly the oil and electricity prices, MUST be presented early on, otherwise the results cannot be interpreted. Only SI Table 1 seems to report "Cost of fossil fuels" as Euros 0.6/L. Presumably these are the jet fuel prices? Why this number, what is the rationale?

Response #1

Thank you for highlighting this important point. We agree that the key assumptions underlying our results, such as kerosene prices and levelized cost of electricity, need to be clearly presented in the main text for better interpretation. In response to your feedback and that of Reviewer 2, we have now included a table at the beginning of the "Scenarios and Framework" section of the results (Table 1, lines 796-798). This table details the key input parameters, including references to the values we chose or the rationale for choosing them. For example, we show that the fossil kerosene price is assumed to be €0.6/l, which is the average jet fuel price between 2021-2023. In addition, we have explicitly mentioned these key assumptions in the text (lines 91-96) and provided a description of the business-as-usual scenario and scenario attributes (cf. Response #2b to Reviewer 2), as shown in lines 90-91. We hope that these additions improve the clarity and understandability of our manuscript.

Comment #2

The critical role of the jet fuel price in that regard can be seen also from Figure 2. The y-axis of this figure is labelled as "costs", whereas the caption states "Cost to achieve CO2 and climate neutrality..." In order to be more meaningful, the authors should subtract the costs of fossil jet fuel use in the baseline scenario in order to illustrate the EXTRA or MARGINAL costs for achieving these targets. Also, shouldn't this figure show cumulative discounted costs instead of year 2050 costs? The magnitude of the numbers of the y-axis suggests this,

but it is not clear from the text. If yes, what is the discount rate?

Response #2

Thank you for this important comment, which was also raised by reviewer 2 (see Response #2a). We agree that presenting annual costs for 2050 may not be very appealing to readers. Therefore, we have decided to present in Figure 2 the cost per liter to achieve CO₂ and climate neutrality through the DAC-based pathways. We decided not to include a figure showing the cumulative cost difference to business-as-usual in the main manuscript because of the challenge of calculating the cost difference for all components (we cannot subtract the cost of fossil fuel from each cost component, only from the total cost). However, we have included a figure in the Supplementary Information (Supplementary Figure 9) showing the cumulative cost difference for different scenarios and referenced it in the main text (lines 134-135). In addition, we have clarified that no discount rate is applied, as shown in the new Table 1, which provides an overview of all key input assumptions.

Comment #3

As partly covered in the above comments, the authors need to distinguish clearly between costs and prices. In the current manuscript, these terms seem to be used interchangeably, particularly wrt "fossil fuels" (do the authors mean "jet fuel"?) and electricity.

Response #3

Thank you for bringing this issue to our attention. We have now made a clear distinction in our manuscript: we refer to "costs" when discussing the increase in flying due to DAC-based mitigation and when highlighting the role of electricity. When discussing fossil kerosene, we use the term "price" and refer to the IATA market price. In addition, we refer to the "increase in the price of air tickets per passenger" because our method of calculation includes the DAC-induced incremental costs added to ticket prices. This should help clarify the distinction between costs and prices throughout the manuscript (e.g., lines 18, 173, 178-179, 181, 184-185, 186, 248, 257, 258, 262, 272, 287).

Comment #4

I also suggest simplifying the multiple colored tables in the text. Which of these variables matter most? Can this be shown graphically?

Response #4

Following your suggestion, we have redesigned the graph to better illustrate the impact of the various variables (Figure 6). In addition, we have implemented an optimization process to minimize cost differences between scenarios by varying one key input variable at a time (lines 242-245, cf. Response #4 to reviewer 2). This allows for a clearer visualization of the variations in input parameters and provides more explicit information compared to the colored tables.

Comment #5

The authors also state that both pathways penetrate the fuels market following S-curves. Can this be shown graphically? I have difficulties imagining the feasibility of adding thousands of large-scale DAC systems and potentially synthetic fuel plants within only 25 years if following an S-curve.

Answer #5

To achieve our ambitious goal of 100% synthetic-fueled or DACCS-based aviation, enormous efforts are indeed required. We discuss the unlikelihood of this in lines 322-324. While

technologies typically follow an S-curve when fully substituting incumbent technologies, we agree that such a rapid ramp-up to meet "externally determined" targets may indeed not have a period of slower growth where substitution rates asymptotically approach 100%. As such, we have adjusted our analysis to show a power curve instead, and provide an explanation in lines 85-87, also highlighting that the ramp up of synthetic fuels assumed in our analysis is very optimistic. The revised diffusion model of DAC is now graphically illustrated and can be seen in Supplementary Figure 2.

Comment #6

My main concern relates to the overarching framework. Presumably as a way to ensure consistency, "All scenarios are based on the same demand for jet fuel" (Methods section). However, the different fuel costs associated with achieving CO₂ or climate neutrality following the two pathways will result in different fares and thus reductions in air transportation demand. This, in turn, will mitigate the demand for fuels along with CO₂ emissions/climate impact, but also result in different losses of consumer welfare. According to my point of view, a rigorous analysis needs to ensure consistency based on fundamental economics. Adding this step would require additional work, but significantly strengthen the economic foundation of the analysis.

Response #6

Thank you very much for your insightful and detailed feedback. We recognize that the lack of dynamic linkages between cost increases and demand reductions is a limitation of our study and have now explicitly stated this in two instances, in the Results (l. 203-206) and in the Discussion (l. 339-342).

We do not go into the broader welfare effects (i.e., including consumer surplus) of different fuel choices, as the reviewer suggests we could, for three reasons. The first is that these will be highly dependent on the policy instrument used to stimulate demand for DACCS and DAC-based synthetic fuels. For example, a blending quota, such as the one being experimented with by the EU, passes on the additional cost of low carbon fuel to consumers, potentially leading to a certain reduction in air travel and consumer surplus. Conversely, a production tax credit, as pioneered in the United States starting in 2025, subsidizes the additional cost of fuel from general government revenues; it does not significantly increase ticket prices, and hence would have a minimal impact on demand.

The second reason is that the response of demand to price increases is not only elastic but also time-dynamic, as consumers may become increasingly willing to pay higher prices for climate-neutral air travel as they experience, for example, the increasingly negative impacts of climate change. Including the price elasticity of demand in our analysis would thus introduce two additional sources of uncertainty due to (i) the varying impact on consumers of policies to achieve aviation climate targets and (ii) the uncertain evolution of the willingness-to-pay for the mitigation of air travel.

The third reason has to do with the objective of our analysis. A consideration of the welfare impacts of higher fuel prices would be extremely relevant within a cost-benefit analysis, in which one is seeking to balance the societal costs of aviation decarbonization with its benefits, in order to determine the efficient level of decarbonization. However, most wealthy countries (and many lower income countries) have adopted net-zero targets, and hence have already committed to eliminating, 100%, the climate impact of aviation; in doing so, they have implicitly accepted the potential loss of consumer surplus (as now discussed in lines 354-366) from achieving climate neutrality, and viewed this as preferable to the risks

associated with continuing net emissions. In this context, cost-benefit analysis becomes less relevant than cost-effectiveness analysis, which focuses on identifying the least costly option to achieve the climate neutrality target. Our paper aims to determine which fuel pathway is the most cost-effective. It is safe to assume that the option with the lowest direct costs will also result in the least loss of consumer surplus, and yet quantifying the latter does not add value to a cost-effectiveness analysis.

For these reasons, we consider this dynamic relationship out of the scope of our analysis and therefore, intentionally, have chosen not to include it. Nevertheless, we consider this demand an exogenous input on which we have performed a sensitivity analysis, recognizing the need for a more thorough assessment of the impact of reducing the rate of growth in demand (l.265-266). Moreover, we recognized that the inclusion of "policy scenarios" in our main analysis was misleading, as these policies in fact have broader impacts on demand and supply that were not adequately accounted for in our framework. For this reason, we have added a new results section on the role of demand reductions (l. 196-235) and eliminated the policy section, with only more qualitative references to the role of policies (l. 277-281). This has led to the modification of the main findings in the Abstract (l. 16-18) and Discussion (l. 288-289).

REVIEWERS' COMMENTS

Reviewer #1 (Remarks to the Author):

I appreciate the authors considering my comments and suggestions, and think that the manuscript is almost ready for publication. I only have minor suggestions that I think would improve the manuscript and clarify some important points:

- Line 19: In the abstract quantify what you mean by “low electricity cost” and “high fossil kerosene prices”. The abstract is one of the most important parts of the manuscript, and it would be useful to state your values early on: “Low electricity cost (XXX) and high fossil kerosene prices (XXX)”
- Lines 25-26: In the following statement I would include that this depends on which global warming potential is used: “About two-thirds of this is attributed to non-CO2 effects, such as contrail cirrus cloud formation or indirect effects due to nitrous oxide emissions”
- Lines 33-34: “could theoretically curb all emissions,” I would actually mention that you are talking about the direct emissions of flying, and not the indirect emissions related to the production of fleets, etc.
- Figure 1 improved a lot, but I still find it a little confusing: the black bars show “net emissions”, but the caption says this represents “uncertainty due to the non-CO2 impacts of aviation”. Which one is it? And if it is net emissions, why are they not net-zero in the climate neutrality scenario? And also: why are there no non-CO2 emissions in the DACCS scenario but there are contrails?
- Lines 118 -119: The title “Emit-and-offset is cheaper under a CO2 neutrality target, but not under a climate neutrality target”, cheaper than what? Than the climate neutrality target? I would state relative to what.
- Lines 136-137: this statement is unclear to me: “Under climate neutrality, where the climate impacts of the two pathways are identical” because figure 1 shows different emissions between scenarios in both carbon neutrality, and climate neutrality. So why do they have identical climate impacts? Do you mean to say that all climate impacts are considered equally?
- Lines 175-177: “Relying exclusively on DAC-based mitigation results in an increase in costs per flight between €23-209 to achieve CO2 neutrality and between €36-329 to achieve climate neutrality” compared to a BAU case? (That is what the caption says). I would also add it in this sentence. And also, shouldn't it be “per passenger flight”? And also, are ranges correct? I see CO2 neutrality costs higher than 250, and climate neutrality costs higher than 400. Am I missing something?
- Line 173 onwards: Another note on section that starts on row 173 is that there are no references to Figure 4a and 4b – which I assume this sections discusses.
- Note for figure 6: the figure talks about changes in +- 70% in parameters, but the caption mentions changes in +- 100%. Which one is it?

- Lines 329-330: “but may be possible if DAC grows at a rate comparable to some historical analogues”, like which one?

Reviewer #2 (Remarks to the Author):

Dear authors and editors,

I've gone through the changes and like them. I have no further comments.

Success with publishing the article. May it be well received and read by a large audience.

Best,

Gerben Hieminga

Reviewer #3 (Remarks to the Author):

The authors have greatly improved their manuscript. I have got only very few minor comments.

1) Eqn 1 on p. 21: Shouldn't Q_t/Q_{t0} be 2 in order for LR to be the learning rate?

2) Same page and SI: The authors state that they calculate the increase in ticket price, which seems to be a typical airfare for a given route plus the cost of DACCX minus the fuel costs. This is not correct, as airlines may not be able to pass on the entire extra costs to passengers. It will depend ultimately on the relative slope of the demand and supply elasticities. The authors should use the direct operating costs rather than the airfare as a basis for this calculation. In the related suppl. table 4, the fuel cost share of 13% for the London to Perth trip seems by far too low.

Can synthetic fuels achieve climate-neutral aviation in a cost-effective way?

We would like to thank the reviewers and the editor for this second round of feedback, and for spotting all these final possible improvements. We are really impressed by the quality of the reviews and believe they have done a great job in improving the manuscript.

In this document, we provide **answers** to all **comments** made by the reviewers. We provide our **answers in blue**.

REVIEWER COMMENTS

Reviewer #1:

I appreciate the authors considering my comments and suggestions, and think that the manuscript is almost ready for publication. I only have minor suggestions that I think would improve the manuscript and clarify some important points:

Many thanks for positive feedback. We implemented as suggested in the comments below, if not stated otherwise.

- Line 19: In the abstract quantify what you mean by “low electricity cost” and “high fossil kerosene prices”. The abstract is one of the most important parts of the manuscript, and it would be useful to state your values early on: “Low electricity cost (XXX) and high fossil kerosene prices (XXX)

Done

- Lines 25-26: In the following statement I would include that this depends on which global warming potential is used: “About two-thirds of this is attributed to non-CO2 effects, such as contrail cirrus cloud formation or indirect effects due to nitrous oxide emissions”

We would like to clarify that this is based on the effective radiative forcing, which is an ‘absolute’ measure of the climate impacts of all these species, and not based on warming potential calculations, which establish a relationship to CO2. To make this clearer, we added this explanation by replacing ‘two-thirds of this’ with ‘two-thirds of aviation’s radiative forcing’.

- Lines 33-34: “could theoretically curb all emissions,” I would actually mention that you are talking about the direct emissions of flying, and not the indirect emissions related to the production of fleets, etc.

Changed to ‘could theoretically curb all *direct flight* emissions’.

- Figure 1 improved a lot, but I still find it a little confusing: the black bars show “net emissions”, but the caption says this represents “uncertainty due to the non-CO2 impacts of aviation”. Which one is it? And if it is net emissions, why are they not net-zero in the climate

neutrality scenario? And also: why are there no non-CO2 emissions in the DACCS scenario but there are contrails?

Thanks for your feedback. To address this source of confusion, we now clarified in the figure caption that the bars represent the uncertainty in the net emissions (the black dots). These uncertainties are only due to the non-CO2 species (other uncertainties are assessed in the sensitivity analysis). Non-CO2 species appear less "visible" in the BAU and DACCS scenarios, because after filtering out the contrail component, the warming and cooling non-CO2 effects almost cancel each other out. However, in the DACCU scenario, the shift to synthetic fuels changes the non-CO2 species, particularly reducing cooling species such as sulfur oxides. This makes the "positive" term more prominent and visible in the graph.

- Lines 118 -119: The title "Emit-and-offset is cheaper under a CO2 neutrality target, but not under a climate neutrality target", cheaper than what? Than the climate neutrality target? I would state relative to what.

We added 'than synthetic fuels'.

- Lines 136-137: this statement is unclear to me: "Under climate neutrality, where the climate impacts of the two pathways are identical" because figure 1 shows different emissions between scenarios in both carbon neutrality, and climate neutrality. So why do they have identical climate impacts? Do you mean to say that all climate impacts are considered equally?

We added 'net climate impacts' to this line. We talk here about the net emissions, which are indeed identical between the two technology pathways for the goal of climate neutrality.

- Lines 175-177: "Relying exclusively on DAC-based mitigation results in an increase in costs per flight between €23-209 to achieve CO2 neutrality and between €36-329 to achieve climate neutrality" compared to a BAU case? (That is what the caption says). I would also add it in this sentence. And also, shouldn't it be "per passenger flight"? And also, are ranges correct? I see CO2 neutrality costs higher than 250, and climate neutrality costs higher than 400. Am I missing something?

You are absolutely correct that we did not fully update the text to reflect the changes in the figure. We have now addressed this, and also incorporated feedback from Reviewer 3 (l. 177-178, 188-190).

- Line 173 onwards: Another note on section that starts on row 173 is that there are no references to Figure 4a and 4b – which I assume this section discusses.

Thanks, added it now.

- Note for figure 6: the figure talks about changes in +- 70% in parameters, but the caption mentions changes in +- 100%. Which one is it?

Thanks for spotting this, indeed the caption was not up to date.

- Lines 329-330: "but may be possible if DAC grows at a rate comparable to some historical analogues", like which one?

Added 'such as wind power'.

Reviewer #2 (Remarks to the Author):

Dear authors and editors,
I've gone through the changes and like them. I have no further comments.
Success with publishing the article. May it be well received and read by a large audience.
Best,
Gerben Hieminga

Many thanks for positive feedback and the wishes.

Reviewer #3 (Remarks to the Author):

The authors have greatly improved their manuscript. I have got only very few minor comments.

Thank you for your positive feedback and recognition of the improvements made to the manuscript. We appreciate your insightful comments and have carefully addressed each of them in the revised version.

1) Eqn 1 on p. 21: Shouldn't Q_t/Q_{t0} be 2 in order for LR to be the learning rate?

Thank you for this question. Indeed, the definition of the learning rate (LR) is 'the decrease in cost due to a doubling of installed capacity'. When the capacity doubles, the exponent simplifies to $1-LR$ because the term $(Q_t/Q_{t0})^{-b}$ simplifies to $1/(1-LR)$ when Q_t/Q_{t0} equals 2. In this case, the corresponding change in cost $(CAPEX_t/CAPEX_{t0} - 1)$ reflects the learning rate.

2) Same page and SI: The authors state that they calculate the increase in ticket price, which seems to be a typical airfare for a given route plus the cost of DACCX minus the fuel costs. This is not correct, as airlines may not be able to pass on the entire extra costs to passengers. It will depend ultimately on the relative slope of the demand and supply elasticities. The authors should use the direct operating costs rather than the airfare as a basis for this calculation. In the related suppl. Table 4, the fuel cost share of 13% for the London to Perth trip seems by far too low.

Thanks for your valuable feedback. We have revised our approach to calculating the impact on ticket prices. Previously, we derived the share of fuel costs by calculating the share of kerosene costs within the typical airfare for the three flights analyzed. However, we recognized the uncertainty this introduced due to variations in profit margins and have now addressed this issue.

In the updated approach, we calculate non-fuel operating costs for the different routes based on the share of kerosene in total operating costs as reported by Ringbeck et al., namely 25% for short-haul flights, 35% for medium-haul flights, and 45% for long-haul flights. In the business-as-usual scenario, we calculate future ticket prices by combining fossil kerosene costs, accounting for efficiency gains, with constant non-fuel operating costs. We also apply a 25% mark-up to reflect the gross profit margin, based on data for U.S. airlines. We then substituted the cost of kerosene with the cost of flying CO₂ and climate neutral with the respective technologies in the DACCS and DACCU scenario, and derived the increase in price relative to the business-as-usual. This way, we did not have to use largely variable airfare prices, which also are obscure in terms of how much gross profit margin they entail.

This revision resulted in changes to the results that are now reflected in Figure 4b and in the manuscript on lines 188-190.